# Supplement Consumption by Elite Soccer Players: Differences by Competitive Level, Playing Position, and Sex

**DOI:** 10.3390/healthcare12040496

**Published:** 2024-02-19

**Authors:** Jaime Sebastiá-Rico, José Miguel Martínez-Sanz, Jesús Sanchis-Chordà, Miguel Alonso-Calvar, Pedro López-Mateu, David Romero-García, Jose M. Soriano

**Affiliations:** 1Area of Nutrition, University Clinic of Nutrition, Physical Activity and Physiotherapy (CUNAFF), Lluís Alcanyís Foundation-University of Valencia, 46020 Valencia, Spain; jaime.sebastia@fundacions.uv.es; 2Food & Health Lab, Institute of Materials Science, University of Valencia, 46980 Paterna, Spain; jose.soriano@uv.es; 3Nursing Department, Faculty of Health Sciences, University of Alicante, 03690 Alicante, Spain; 4Area of Nutrition, Academia Valencia CF SAD, 46980 Paterna, Spain; 5Area of High Conditional Performance, Academia Valencia CF SAD, 46980 Paterna, Spain; 6Area of Medical Services, Academia Valencia CF SAD, 46980 Paterna, Spain; 7Faculty of Health Sciences, University of Alicante, 03690 Alicante, Spain; 8Joint Research Unit of Endocrinology, Nutrition and Clinical Dietetics, Health Research Institute La Fe-University of Valencia, 46026 Valencia, Spain

**Keywords:** ergogenic aids, football, performance, soccer, sports nutrition, supplementation

## Abstract

Soccer is a sport practiced all over the world and whose practice begins in young athletes. Currently, the consumption of nutritional supplements is essential to achieve the maximum performance of players. The aim of this study was to describe the consumption of sports supplements (CSS) by elite soccer players and its association with their competitive level, playing position, and sex. A comparative descriptive and non-experimental study was performed during the 2021–2022 competitive season. A total of 70 elite players completed one online questionnaire about their CSS. We found that sports drinks (55.7%), sports bars (50.0%), whey protein (48.6%), caffeine (47.1%), and creatine (60.0%) were the most consumed supplements by the total sample. Relative to the categories, the CSS was higher in the senior teams for both men and women. Regarding playing positions, caffeine was more consumed by midfielders and forwards (*p* = 0.013). Finally, in relation to sex, significant differences were found in the consumption of sports confectionery (*p* = 0.036), whey protein (*p* = 0.002), β-alanine (*p* = 0.013), and melatonin (*p* = 0.016). Soccer club SS questionnaires gather data to understand patterns, assess effectiveness and risks, and aid research. In conclusion, differences were found in the CSS according to competitive level, sex, and playing position.

## 1. Introduction

Soccer is a physically demanding intermittent sport that requires high physical and metabolic demands [1], recognized as one of the most popular and practiced sports worldwide. The game involves two teams of 11 players, in both men’s and women’s soccer, competing against each other to score goals [2]. There are two main categories: the base and the professional. The base category (U19-10) is played by young people before reaching the professional leagues, such as the First and Second Divisions in Spain [2]. It is widely recognized that an athlete’s physical performance is associated with their body composition [3], hydration, healthy eating habits, and supplementation [4,5,6,7]. Understanding how playing position and other variables can impact the nutrition habits of soccer players is crucial for creating effective nutrition education programs [5].

In recent years, supplements have become increasingly popular in the sports industry due to their ability to enhance performance, especially in energy-intensive sports where dietary intake alone may not be sufficient. However, the supplement industry is driven by economic motives and responds to consumer demand, so there is sometimes a lack of evidence for the consumption of certain supplements and there may be industry-driven publication or advertising biases [5]. The detection and prevention of contamination of ergogenic supplements with banned substances is crucial to ensure the safety of athletes. Without proper control in the selection of supplements, there is a risk of adverse effects [8].

In 2000, the Australian Institute of Sport (AIS) established a nutritional supplement classification system (known as the ABCD system) for athletes. This system differentiates supplements according to the level of existing scientific evidence, as well as other parameters related to safety, legality, and effectiveness in improving sports performance [9]. Therefore, supplements belonging to group A are those that have a high level of scientific evidence to improve performance. These are divided into sports foods (specialized products are utilized to offer a convenient nutrient source in situations where consuming regular foods is not feasible, such as sports drinks, bars and gels, and whey protein), medical supplements (supplements used to prevent or treat clinical issues including diagnosed nutrient deficiencies, such as iron, calcium, zinc, and probiotics), and performance supplements (supplements/ingredients that can support or enhance sports performance, such as caffeine, B-alanine, creatine, sodium bicarbonate, nitrates, and glycerol) [9]. Group B includes supplements that may have a positive effect on certain conditions, although further research is needed (e.g., menthol, curcumin, vitamin C, and fish oils), while there is insufficient scientific evidence to support the benefits of Group C supplements among athletes, or no research has been conducted to guide an informed opinion (e.g., magnesium, vitamin E, leucine, and phosphate). Lastly, group D includes those supplements banned or at high risk of contamination with substances that could lead to a positive doping test, so they cannot be used by athletes (e.g., higenamine, ostarine, sibutramine, and androstenedione) [9].

Several studies [4,10], along with two consensus statements, one from the Portuguese Football Federation [6] and another from the Union of European Football Associations (UEFA) expert group [5], have highlighted the consumption of different supplements aimed at increasing or improving sports performance. Despite the growing popularity and scientific advancements in women’s soccer, research on female players in this sport remains significantly lower compared to men’s soccer [5]. Furthermore, the use of supplements in adolescent athletes is a controversial topic. While some argue that consumption can enhance performance and aid in recovery during the competitive season, there are potential risks associated with their use. These risks include lack of regulation, possible adverse side effects, and the presence of banned substances [8,11].

Accordingly, the aim of this study was to describe the consumption of sports supplements (SS) by elite soccer players and its association with their competitive level, playing position, and sex. The initial hypothesis was that:

**Hypothesis 1 (H1).** 
*Higher category soccer players will have higher consumption of SS.*


**Hypothesis 2 (H2).** 
*Male and female soccer players will have similar SS intake, but not SS type.*


**Hypothesis 3 (H3).** 
*The amount and type of SS consumed will be similar in the different playing positions.*


## 2. Materials and Methods

### 2.1. Type of Study

A descriptive, cross-sectional, and non-experimental study was conducted to determine the consumption and regular use of dietary supplements by elite soccer players of both sexes attending the Valencia C.F. Academy. The assessment was made in the month of May during the competition season 20–21. Sample size calculation was performed using Rstudio software (version 3.15.0, Rstudio Inc., Boston, MA, USA). The significance level was set a priori at *p* = 0.05. The standard deviation (SD) was set according to the total SS data from previous studies in Spanish elite athletes (DE = 2.1) [10]. With an estimated error (d) of 0.49, the required sample size was 70 athletes. The study population was selected by non-probability, non-injury, convenience sampling among elite soccer players of both sexes attending the Valencia C.F. Academy.

### 2.2. Participants

The SS consumption questionnaire included templates from 100% of Valencia Mestalla (n = 21), Juvenil A (n = 22), Juvenil B (n = 23), Valencia CF Femenino (n = 15), and Valencia CF Femenino B (n = 13). However, only 70 out of 94 soccer players (74.5%) completed the voluntary questionnaire, including 12 Valencia Mestalla players, 15 Juvenil A players, 15 Juvenil B players, and all players from Valencia CF Femenino (15) and Valencia CF Femenino B (13). All players had at least 9 years of soccer training experience and performed from 4 up to 7 regular training sessions per week (approximately 90 to more than 120 min per day), playing a theoretical official soccer match per week. The criteria for inclusion in this study were as follows: (a) be a healthy subject with medical authorization for the practice of federated sport; (b) belong to a team of Valencia C.F. Academy; (c) being federated in soccer; and (d) training a minimum of 4 days per week. The exclusion criteria for this study were as follows: (a) having been injured or having become ill during this study. The study population overlaps with that of a previous study published in Nutrients [7].

### 2.3. Procedure

In order to select the sample, the C.F. Valencia Academy sent out a statement informing local and non-resident players about the study, instructions, and an invitation to participate. Before the players completed the questionnaire, they were informed of the purpose of the study. Informed consent was obtained and signed by those responsible for the study, as well as the medical and coaching staff of the Valencia C.F. Academy. It was also signed by each participant and their respective parents or legal guardians. The research team was present at the Valencia CF sports center while the players filled out the questionnaires in order to clarify any doubts.

The protocol conforms to the tenets of the Declaration of Helsinki for research involving human participants and has been approved by the Ethics Committee of the University of Valencia (1534145).

### 2.4. Instruments

This study used a questionnaire that had previously been used in similar studies [12,13]. The selected questionnaire about the consumption of sports supplements (CSS) was validated for content, applicability, structure, and presentation by Sánchez-Oliver [14]. This questionnaire was developed by a team of three sports scientists and 25 experts in sports science, sports medicine, nutrition, chemistry, and pharmacology to ensure its construction validity [14]. In fact, in a study conducted by Knapik et al. [15], who analyzed the quality of questionnaires designed to determine the prevalence of SS use by athletes, this form obtained a methodological score of 54%. It stood out as one of the 57 questionnaires reviewed (out of a total of 164) that was considered adequate to collect accurate information on the SS used by athletes. The questionnaire consists of 28 questions divided into three main sections. The first section collects the subject’s age, team (which identifies the subject’s gender), and playing position. This section has 3 questions. The second section includes 7 questions that collect the sport practice and its context, such as years of practice and number of weekly training sessions. The third section includes 4 questions focused on nutrition, such as whether he/she follows any type of diet and who advises him/her on nutrition. The last section, the most extensive, focuses on diet and supplement consumption. It contains 14 questions, such as what supplements the respondents used, why they used them, who advised them to use them, where they bought them, when they took them, and their perception of the results after use. The questionnaire can be obtained by contacting the authors via email.

### 2.5. Stadistical Analysis

The Kolmogorov–Smirnov test was applied to check whether the variables had a normal distribution and the Levene test was used to verify homoscedasticity. For the descriptive analysis, the mean and standard deviation (M ± SD) were used. An ANOVA analysis was performed according to sex (male-female), category (senior-base), and positions (goalkeeper, defense, midfielder, and forward) to analyze the differences in the SS consumed from the different categories determined by the AIS. For supplements that were consumed by more than 10% of the sample, a chi-square test (χ^2^) was performed to verify possible differences according to sex, category, and positions. The level of statistical significance was set at *p* < 0.05. Statistical analysis was performed with the Statistical Package for Social Sciences v.20 (SPSS).

## 3. Results

Table 1 shows the characteristics of the players (age, training days, and extra training days) by gender, competition level, and position.

Table 2 shows the mean with its standard deviation and the ANOVA of the age, training days, and SS consumed of the segmented sample according to sex and category. Regarding sex, significant differences were found in age (F = 5.402; *p* = 0.023), with greater age being observed in women, while for the rest of the variables no significant differences were found (F = 0.020–2.885; *p* = 0.889–0.094). Regarding the category, significant differences were found for age (F = 111.883; *p* = 0.000) and days of training (F = 12.676; *p* = 0.001), both in the total and in all of the supplements of group A (F = 4.531–8.781; *p* = 0.037–0.004) and in the supplements of group C (F = 9.693; *p* = 0.003), observing in all cases higher values in the senior category.

Table 3 shows the mean with its standard deviation and the ANOVA of the age, training days, and SS consumed of the segmented sample according to playing position. All players trained an average of 5 days and 2 extra days, with the exception of the goalkeepers who trained 1 day more on average. Regarding the consumption of supplements, it was observed that, in all positions, sports foods belonging to group A supplements were the most consumed, while group D supplements were the least consumed. It was also observed that in all positions the supplements belonging to group C were more consumed than those belonging to group B. And regarding the differences between the positions, no significant differences were found for any of the analyzed variables (F = 0.342–2.424; *p* = 0.795–0.074).

Table 4 shows those supplements that were consumed by more than 10% of the total sample and according to sex, category, and playing positions. It was observed that sports drinks (55.7%), sports bars (50.0%), whey protein (48.6%), caffeine (47.1%), and creatine (60.0%) were the most consumed supplements by the total sample. Regarding sex, significant differences were found in the consumption of sports confectionery (*p* = 0.036), whey protein (*p* = 0.002), β-alanine (*p* = 0.013), and melatonin (*p* = 0.016), observing a higher consumption in women, with the exception of sports confectionery, which was consumed more by men. Regarding the category, significant differences were found in whey protein (*p* = 0.016), carbohydrate gainers (*p* = 0.011), β-alanine (*p* = 0.001), leucine (*p* = 0.004), and melatonin (*p* = 0.001), observing a higher consumption in all of these cases in the senior category. And finally, with respect to playing positions, significant differences were only found for caffeine consumption (*p* = 0.013), where midfielders and forwards had higher values.

Of the 70 participants, 42 (60.0%) followed some type of diet, with the Mediterranean diet being the most popular (83.3%) followed by the flexible diet (9.52%) and intermittent fasting (7.18%). The main reasons for dieting were to improve performance (71.5%), to take care of their health (26.5%), and to treat injuries (2.3%). Of all dieters, they were mainly advised by the club’s internal nutritionist (80.9%) or by a dietitian/nutritionist external to the club (9.55%), although some of them did not seek advice (9.55%).

In relation to the opinion of the consumption of SS, 62 soccer players were in favor of their consumption (88.5%), while 6 did not answer (8.6%) and 2 were not in favor (2.9%). However, most soccer players have consumed SS at some stage in their career (91.5%). In relation to the intake of SS, soccer players mainly consume them both in training and on match days (78.5%), especially after sports practice (60.0%), although there were 20 players who ingested them before, during, and after sports practice (28.6%), and 8 who only consumed them before sports practice (11.4%).

The main reasons for consuming these SS were to improve sports performance (80.0%), to take care of health (27.1%), and to improve physical appearance (20.0%). Soccer players obtained supplements mainly through the club itself (94.2%), but also through the Internet (14.2%), pharmacies (7.1%), and herbalists (7.1%); being motivated to take them especially by the club’s internal nutritionist (65.7%), physical trainer (18.6%), external club dietitian/nutritionist (11.4%), and doctor (4.3%).

## 4. Discussion

To our knowledge, this is the first study to analyze the differences in SS consumption among elite soccer players by different categories, playing positions, and genders. The main findings of this study were: (1) senior soccer players were more likely to consume SS; (2) although similar patterns were observed between sexes, differences emerged in the consumption of sports confectionery, whey protein, β-alanine, and melatonin; (3) only differences in caffeine consumption were highlighted among the different playing positions, with higher levels among midfielders and forwards. 

### 4.1. Influence of Competitive Level, Playing Position, and Sex

It is increasingly common to see athletes taking supplements to improve their performance, health, aesthetics or for other reasons such as sponsorship or advertising [16,17]. In our study, 88.5% of the soccer players were in favor of SS consumption and at least 91.5% had consumed them at some time, so this is a population susceptible to the consumption of these products. However, nutritionists were primarily responsible for advising on the consumption of SS in our soccer player sample, with a 77.1% presence of both internal and external club nutritionists. Sports clubs should have nutritionists oversee the implementation of dietary and SS recommendations and provide education on the subject. These professionals should be qualified to critically evaluate the translational potential of research and the applicability of these results in practice [18,19,20]. It is worth noting that soccer players who followed a dietary strategy were often advised by a nutritionist, with 90.45% of cases involving both internal and external nutritionists. This is similar to the advice provided about SS in our work and other sports [12,13,21].

In addition, the main motive for consumption was the pursuit of sports performance, followed by health care and improving physical appearance. This last motive is important to mention, as demanding a physique with a high lean mass and a very low fat mass can be detrimental to performance and increase the risk of injury [5,22]. Like athletes in any other sport, soccer players are not immune to developing and suffering from eating disorders such as anorexia, orthorexia, and vigorexia. These disorders can harm not only sports performance but also the athlete’s health [23]. Excessive perfectionism is a key psychopathological trait that underlies the development of eating disorders in soccer players, despite the multifactorial nature of the pathology [24]. In fact, the use of Lycra sportswear has been one of the factors driving the interest in body aesthetics over performance [25].

In relation to scientific studies, there are several studies whose results show that the SS most consumed by athletes are caffeine, vitamins and minerals, creatine, antioxidants, and protein supplements [16,26,27,28]. In our study, sports drinks and sports bars were also two of the most consumed supplements by the sample, something that was also observed in the “División de Honor” soccer players [26]. Scientific studies have shown a correlation between carbohydrate intake from food and SS, such as sports drinks and bars, and improved performance in soccer [29,30,31,32,33]. Therefore, it is understandable that consumption of these supplements was high in our sample of soccer players at all competitive levels, sexes, and playing positions. As mentioned above, caffeine and creatine are two of the most consumed SS by athletes, both of them being supplements classified by the AIS in category A, focused on improving sports performance [34]. These two supplements are popular among soccer players [5,6,35] as they are both used to improve performance and are considered legal. 

Creatine is a non-essential organic nitrogenous compound from the amine group that increases muscle creatine reserves, leading to a higher rate of phosphocreatine resynthesis [36]. This has been well demonstrated in soccer, as it translates into improved performance, particularly in high-intensity intermittent efforts [37,38,39,40,41]. Caffeine is an alkaloid of the methylated xanthine family and is considered a stimulant with well-established benefits for athletic performance and decreased fatigue [36,42]. In addition to its effects on alertness and fatigue during physical exertion, caffeine has also been found to enhance soccer-specific skills such as jumping, sprinting, agility, ball control, and passing accuracy. This explains its widespread use among athletes and its popularity in scientific research [5,6,43,44,45,46,47,48,49].

Regarding women’s soccer, there are few studies that have examined the prevalence of SS use. A recent study evaluated the use of dietary supplements in 103 elite soccer players, with 82% reporting use within the last 12 months, a result consistent with our findings [50]. However, the main reason for consuming these products was to take care of their health (66.0%) and they were mainly advised by physicians (46.0%), followed by dieticians/nutritionists. The most commonly used supplements were vitamin D (52.0%), omega-3 fatty acids (49.0%), and protein powder (45.0%). In our study, while whey protein was also frequently used (71.4%), the consumption of Vitamin D (25.0%) and omega-3 fatty acids (25.0%) was not prominent. However, creatine, caffeine, sports drinks, and sports bars were popular choices. While there is a high incidence of vitamin D deficiency among soccer players [51,52] and vitamin D supplementation has been extensively studied in this population [53,54,55,56], we hypothesize that, in most cases, the players in our study do not require vitamin D supplements due to the favorable weather conditions in the region where they train and compete, regular monitoring of their blood levels through periodic clinical analysis, and dietary advice from the club’s nutritionist or, in some cases, an external dietician/nutritionist. This last reason could also justify the low consumption of omega-3 fatty acids through supplements, since with well-structured dietary guidelines by a dietician/nutritionist and the absence of sports injuries or pathologies, supplementation is not necessarily required [57,58]. 

Comparing our own results between sexes, the consumption of melatonin (28.6% vs. 7.1%) and β-alanine (25.0% vs. 4.8%) was higher than in men, although men consumed more sports confectionery (21.4% vs. 3.6%). β-alanine is classified by the AIS in category A for sports performance enhancement [34]. It is a nonessential amino acid synthesized in the liver and acts as a limiting precursor for carnosine dipeptide synthesis in human skeletal muscle, as opposed to L-histidine [36]. Baseline intramuscular carnosine levels are 12–72% lower in women compared to men [59,60]. Additionally, it has been observed that females may require lower levels of β-alanine supplementation to achieve similar relative increases in carnosine compared to males [60]. Although the evidence in female soccer players is inconclusive [61,62], there is a strong scientific rationale for supplementing with β-alanine in both men and women [4,5,10,63,64]. 

Melatonin is a peptide hormone commonly used as a nutritional supplement to regulate the sleep/wake cycle. However, it also has other physiological effects that can improve athletic performance [36]. In soccer, melatonin is often used to improve sleep hygiene and mitigate the negative effects of jet lag after international travel [5]. Several studies have evaluated the impact of melatonin on elite male soccer players, both adult and youth, with promising results [65,66,67,68,69,70,71]. However, there is a lack of research on the effects of melatonin on female soccer players. The authors suggest that female soccer players may consume more melatonin due to a need for better quality rest and higher levels of daily stress caused by personal, social, cultural, and economic conditions that differ from those of male soccer players. In many cases, female soccer players need to balance sports competitions with work or study, resulting in increased stress levels [72].

The use of supplements in adolescent athletes is a contentious issue. First, certain supplements may be beneficial for improving athletic performance and optimizing muscle recovery [5]. Nevertheless, it is important to keep in mind that adolescents are still developing and their bodies have specific nutritional needs that can be met by a balanced diet [73]. In our work, we have found that the consumption of SS is less frequent among youth categories compared to senior categories. However, it is important to consider the dietary needs of young soccer players, such as calcium, iron, and vitamin D, as their requirements differ from those of adults [5,74]. Therefore, if there is a deficiency in these nutrients, supplementing them would be a beneficial intervention by the club’s medical team. This should be performed in addition to interventions focused on performance enhancement, as long as the doses and forms of administration are appropriate for the young athlete [5,75].

Research suggests that during a match, the midfielder and forward positions are the most physically active playing positions [76,77,78]. Furthermore, research has shown that these positions are associated with lower levels of body fat [79], indicating a higher energy expenditure. Our study found a positive association between these positions and a higher caffeine intake. However, the club has a budget for the provision of sports supplements and the support of dietitians/nutritionists. Therefore, it is reasonable to assume that there are no significant differences in the amount and type of supplements consumed, as they are mostly sports foods suitable for any soccer player [5,6,34]. To our knowledge, the present study is the first to analyze the consumption of dietary supplements by elite soccer players, differentiating by competitive level, gender, and playing position (Figure 1).

Finally, SS consumption may vary among teams based on the club’s budget. However, as a general rule, men’s and higher category squads have a larger budget than lower category or women’s squads. Additionally, supplement consumption may be recommended by the club’s specialized sports nutritionist based on scientific evidence and tailored to the individual’s competitive level, gender, age, and budget. 

### 4.2. Limitations

The current study presents important information regarding the prevalence and usage of SS in elite soccer. However, there are limitations that must be considered to improve its applicability. Firstly, the sample size in each category was small and there was heterogeneity between groups. Nevertheless, a significant sample size was used in this type of population according to the statistical principles applied. However, the authors caution against generalizing the findings to other sports populations due to the limitations of the data. This study achieved its objective of describing the supplementation habits and practices of elite soccer players of both sexes, which has been little studied so far. Additionally, the information on supplement use was collected through self-reported and retrospective methods, relying on the athletes’ memory. This could lead to errors in the number and type of SS reported. However, in general, athletes tend to be knowledgeable about their dietary habits and concerned about their diet and training, as their performance depends on it. Additionally, supplementation is primarily overseen by the club nutritionist and the coaching and medical staff to promote optimal athletic performance and health in these athletes. Though the CSS is a validated tool and proven to be efficient in elucidating the behavioral patterns of the sampled population, it bears intrinsic limitations in the precise quantification of supplement usage, both in the context of training and competitive events. In addition, the questionnaire does not consider the doses or protocols of the SS consumed, nor does it ask about the brand of supplements. Therefore, it is not possible to estimate the energy and nutritional intake derived from SS. This limitation is present in all questionnaires regarding SS consumption frequency [12,13].

Lastly, this survey was conducted at a single point in the season. Nevertheless, the decision was made to conduct it in the middle of the season so that the soccer players would already be familiar with the consumption of SS. This makes it less likely that they would forget what kind of supplements they have been consuming. Soccer players may exhibit shifts in their supplementation habits over time, owing to alterations in nutritional requirements, evolving professional recommendations, or the influence of dietary supplement trends. The CSS may not comprehensively capture such dynamic variations and might require periodic updates to accurately reflect alterations in consumption trends.

### 4.3. Practical Applications

SS consumption questionnaires can be used in soccer clubs to collect data on the types and amounts of supplements consumed by soccer players. This provides useful information on SS consumption patterns and may help assess the effectiveness and perception of supplements used by elite soccer players. The use of a questionnaire can reveal risky practices related to the consumption of supplements in elite soccer players, such as inappropriate doses or non-recommended combinations of supplements due to possible interactions, or over-reliance on them. This information is crucial in promoting responsible use of supplements among soccer players. Additionally, the questionnaire allows for the collection of data for scientific research in the field of sports supplementation. 

## 5. Conclusions

In summary, a substantial portion of the sample regularly includes various SS in their routine, including sports drinks, energy bars, caffeine, creatine, and whey protein. Notably, this consumption is more prevalent among individuals in higher categories, reflecting the growing interest among athletes in enhancing their capabilities. Nevertheless, it is essential to underscore the necessity for further research focused on supplement use among younger populations. Furthermore, when it comes to the quantity of supplements consumed, both sexes exhibited similar patterns, but distinctions emerged in the consumption of sports confectionery, whey protein, β-alanine, and melatonin. Additionally, differences were observed only in caffeine consumption among playing positions, with midfielders and forwards displaying higher levels.

## Figures and Tables

**Figure 1 healthcare-12-00496-f001:**
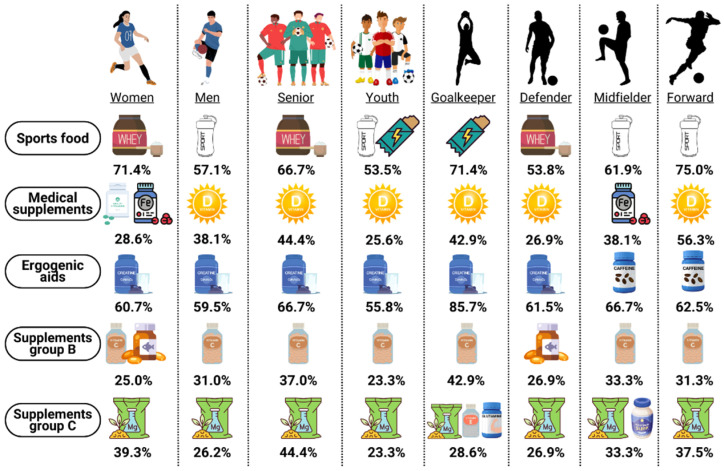
Most used supplements according to competitive level, gender, and playing position.

**Table 1 healthcare-12-00496-t001:** Descriptive data on age and training days as a function of sex, category, and playing positions.

Variable	Sex (Media ± DE)		Category (Media ± DE)		Playing Position (Media ± DE)			
	Men (n = 42)	Women (n = 28)	Senior (n = 27)	Youth (n = 43)	Goalkeepers (n = 7)	Defenders (n = 26)	Midfielders (n = 21)	Forwards (n = 16)
Age	20.02 ± 2.87	22.04 ± 4.38	24.44 ± 3.45	18.56 ± 0.98	21.57 ± 3.87	19.73 ± 3.25	21.76 ± 3.88	21.06 ± 3.77
Training days	5.62 ± 0.76	5.43 ± 0.96	5.96 ± 0.52	5.28 ± 0.91	5.14 ± 0.90	5.50 ± 0.86	5.62 ± 0.86	5.69 ± 0.79
Extra-training days	2.60 ± 1.43	2.21 ± 1.52	2.30 ± 0.95	2.53 ± 1.72	3.43 ± 2.15	2.46 ± 1.45	2.29 ± 1.27	2.19 ± 1.37

**Table 2 healthcare-12-00496-t002:** Descriptive data and ANOVA of age, training days, and SS consumed in the different categories established by the AIS, according to teams.

Variable		Sex (Media ± DE)				Category (Media ± DE)			
		Men (n = 42)	Women (n = 28)	F	*p*	Senior (n = 27)	Youth (n = 43)	F	*p*
Age		20.02 ± 2.87	22.04 ± 4.38	5.402	0.023	24.44 ± 3.45	18.56 ± 0.98	111.883	0.000
Training days		5.62 ± 0.76	5.43 ± 0.96	0.850	0.360	5.96 ± 0.52	5.28 ± 0.91	12.676	0.001
Extra-training days		2.60 ± 1.43	2.21 ± 1.52	1.129	0.292	2.30 ± 0.95	2.53 ± 1.72	0.433	0.513
Supplements Group A	Sports food	2.02 ± 1.44	2.50 ± 1.62	1.661	0.202	2.78 ± 1.72	1.86 ± 1.28	6.508	0.013
	Medical supplements	0.95 ± 1.46	1.11 ± 1.34	0.220	0.641	1.48 ± 1.55	0.72 ± 1.12	5.654	0.020
	Ergogenic aids	1.36 ± 1.10	1.32 ± 0.94	0.020	0.889	1.67 ± 0.83	1.14 ± 1.10	4.531	0.037
	Total	4.33 ± 3.25	4.93 ± 3.14	0.579	0.449	5.93 ± 3.22	3.72 ± 2.91	8.781	0.004
Supplements Group B		0.69 ± 0.87	0.79 ± 1.20	0.149	0.701	1.00 ± 1.27	0.56 ± 0.76	3.306	0.073
Supplements Group C		1.31 ± 1.97	2.25 ± 2.66	2.885	0.094	2.70 ± 2.92	1.05 ± 1.53	9.693	0.003
Supplements Group D		0.02 ± 0.15	0.00 ± 0.00	0.663	0.418	0.04 ± 0.19	0.00 ± 0.00	1.607	0.209

DE: Standard deviation.

**Table 3 healthcare-12-00496-t003:** Descriptive data and ANOVA of age, training days and SS consumed in the different categories established by the AIS, according to teams.

Variable		Playing position (Media ± DE)					
		Goalkeepers (n = 7)	Defenders (n = 26)	Midfielders (n = 21)	Forwards (n = 16)	F	*p*
Age		21.57 ± 3.87	19.73 ± 3.25	21.76 ± 3.88	21.06 ± 3.77	1.375	0.258
Training days		5.14 ± 0.90	5.50 ± 0.86	5.62 ± 0.86	5.69 ± 0.79	0.748	0.527
Extra-training days		3.43 ± 2.15	2.46 ± 1.45	2.29 ± 1.27	2.19 ± 1.37	1.307	0.279
Supplements Group A	Sports food	2.14 ± 1.07	1.92 ± 1.26	2.52 ± 1.96	2.31 ± 1.45	0.624	0.602
	Medical supplements	1.14 ± 1.46	0.73 ± 0.87	1.24 ± 1.81	1.13 ± 1.26	0.625	0.601
	Ergogenic aids	1.57 ± 0.53	0.92 ± 0.80	1.62 ± 1.24	1.56 ± 1.09	2.424	0.074
	Total	4.86 ± 2.12	3.58 ± 2.42	5.38 ± 4.09	5.00 ± 3.20	1.428	0.242
Supplements Group B		0.86 ± 0.69	0.58 ± 0.99	0.86 ± 1.15	0.75 ± 1.00	0.342	0.795
Supplements Group C		1.29 ± 1.38	1.27 ± 2.15	2.38 ± 2.87	1.63 ± 1.96	0.998	0.399
Supplements Group D		0.00 ± 0.00	0.04 ± 0.20	0.00 ± 0.00	0.00 ± 0.00	0.553	0.648

DE: Standard deviation.

**Table 4 healthcare-12-00496-t004:** Distribution (%) of the most consumed supplements by position according to the categories established by the AIS.

Category		Supplement	Total (%)	Sex (%)			Category (%)			Playing position (%)				
				Women	Men	*p*	Senior	Youth	*p*	Goalkeepers	Defenders	Midfielders	Forwards	*p*
Supplements Group A	Sports food	Sports drinks	55.7	53.6	57.1	0.768	59.3	53.5	0.636	42.9	42.3	61.9	75.0	0.165
		Sports confectionery	14.3	3.6	21.4	0.036	18.5	11.6	0.423	14.3	3.8	19.0	25.0	0.240
		Sports bars	50.0	42.9	54.8	0.329	44.4	53.5	0.461	71.4	50.0	47.6	43.8	0.663
		Whey protein	48.6	71.4	33.3	0.002	66.7	37.2	0.016	42.9	53.8	42.9	50.0	0.881
		Carbohydrate gainers	15.7	25.0	9.5	0.081	29.6	7.0	0.011	28.6	7.7	23.8	12.5	0.348
	Medical supplements	Iron	21.4	28.6	16.7	0.234	33.3	14.0	0.054	28.6	7.7	38.1	18.8	0.084
		Vitamin D	32.9	25.0	38.1	0.253	44.4	25.6	0.102	42.9	26.9	19.0	56.3	0.089
		Multivitamin	21.4	28.6	16.7	0.234	33.3	14.0	0.054	28.6	15.4	28.6	18.8	0.687
		Multimineral	11.4	14.3	9.5	0.540	11.1	11.6	0.947	14.3	7.7	19.0	6.3	0.563
		Probiotics	11.4	10.7	11.9	0.878	18.5	7.0	0.140	0.0	15.4	9.5	12.5	0.706
	Ergogenic aids	Caffeine	47.1	42.9	50.0	0.558	63.0	37.2	0.036	42.9	23.1	66.7	62.5	0.013
		Β-alanine	12.9	25.0	4.8	0.013	29.6	2.3	0.001	28.6	3.8	19.0	12.5	0.246
		Creatine	60.0	60.7	59.5	0.921	66.7	55.8	0.367	85.7	61.5	52.4	56.3	0.465
Supplements Group B		Omega 3	24.3	25.0	23.8	0.909	29.6	20.9	0.409	14.3	26.9	23.8	25.0	0.922
		Vitamin C	28.6	25.0	31.0	0.589	37.0	23.3	0.214	42.9	19.2	33.3	31.3	0.552
		Carnitine	11.4	14.3	9.5	0.540	14.8	9.3	0.480	28.6	7.7	19.0	0.0	0.129
Supplements Group C		Essential amino acids	12.9	17.9	9.5	0.308	22.2	7.0	0.064	0.0	15.4	19.0	6.3	0.471
		Magnesium	31.4	39.3	26.2	0.248	44.4	23.3	0.063	28.6	26.9	33.3	37.5	0.901
		Leucine	14.3	14.3	14.3	1.000	29.6	4.7	0.004	14.3	11.5	19.0	12.5	0.899
		Vitamin E	11.4	7.1	14.3	0.357	11.1	11.6	0.947	28.6	7.7	14.3	6.3	0.394
		Melatonin	15.7	28.6	7.1	0.016	33.3	4.7	0.001	0.0	7.7	33.3	15.7	0.055
		Glutamine	18.6	21.4	16.7	0.616	25.9	18.6	0.210	28.6	15.4	23.8	12.5	0.704

## Data Availability

The data presented in this study are available in the tables of this article. The data presented in this study are available on request from the corresponding author.

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
