# Peer review of "Supplement Consumption by Elite Soccer Players: Differences by Competitive Level, Playing Position, and Sex"

_healthcare, 2024, doi:10.3390/healthcare12040496_

Round 1

Reviewer 1 Report

Comments and Suggestions for Authors

The study focuses on Identifying sports supplements (SS) intake pattern in elite soccer players and aims to understand its association with gender, playing position and competitive level. Study offers unique insight in rarely addressed are of supplemental intake in female athletes.

The authors have tackled a relevant subject and provide evidence overall SS consumption and certain types of CSS. Please consider the following suggestions to improve the impact of the article.

1.        Hard to review as the format doesn’t seem to follow traditional guidelines. Line numbers are vital to review a paper.

2.        Page 1 and 2 – Introduction started of appropriately, however lacks relevancy. Recommend authors incorporate studies that potentially identify with the same topic along with supplement classification information by AIS. There is lack of evidence that leads to last statement “However, there are few studies that focus solely on supplementation in the younger and female population and no definitive conclusions can be drawn.” Do the authors suggest there is sufficient evidence in male population? If so need to highlight this.

3.        Page 3 – Recommend moving descriptive data (participant characteristics) to the results section. Under the participants section, describe participants in detail and include inclusion and exclusion criteria.

4.        Page 4 – Results section can begin with descriptive data highlighting participant characteristics.

5.        Page 6 – Reword “Table 3 shows the mean with its standard deviation and the ANOVA of the age, training days and SS consumed of the segmented sample according to playing position. No significant difference was found in any of the variables analyzed (F = 0.342 - 2.424; p = 0.795 - 0.074).” The above statement can be reworded to incorporate more details. AS it, represents a table/figure legend.

6.        Page 10: Discussion – Recommended replacing “unintentional doping” to a more scientific terminology. Or simple stating “ There is risk of adverse effect”

7.        Page 10: “However, the nutritionist is the main person in charge of advising on the consumption of SS in the sample of our soccer players, with a presence of 77.1% combining the club's internal nutritionist (65.7%) as well as external ones (11.4%)” Statement is unclear. Is the nutritionist promoting SS use or advising about SS consumption in general? Clarity is needed on this statement.

8.        The statement that follows is also unclear – “This is important, as this is one of the figures suitable for recommending the consumption of supplements, as they have scientific knowledge in the field of nutrition and supplementation, which implies a better choice by athletes when opting to consume some SS or others in terms of their level of scientific evidence and safety [16–18].” Clarity is needed – whether nutritionist is advising on safe choices or confirming SS use with Athletes. Was education piece assessed in this study? Results were not reported for this. The statement is very vague.

9.        Since hypotheses were provided earlier, the discussion section can begin if these were accepted or rejected. This way it becomes easier for the reader to understand the discussion points. 

Comments on the Quality of English Language

There is an average command of English language in general. Some sentences seem very vague especially in the discussion section. The results section is very brief and discussion points do not align well with the results. Overall there is extensive use of contrast and prepositions though out the text making the paper come across very layman like as opposed to scientific writing. Would recommend proofreading the whole article and revising as needed to improve the quality.

Author Response

The study focuses on Identifying sports supplements (SS) intake pattern in elite soccer players and aims to understand its association with gender, playing position and competitive level. Study offers unique insight in rarely addressed are of supplemental intake in female athletes.

The authors have tackled a relevant subject and provide evidence overall SS consumption and certain types of CSS. Please consider the following suggestions to improve the impact of the article.

  1. Hard to review as the format doesn’t seem to follow traditional guidelines. Line numbers are vital to review a paper.

Response of the authors: Following the reviewer's suggestions, line numbers were added.

  1. Page 1 and 2 – Introduction started of appropriately, however lacks relevancy. Recommend authors incorporate studies that potentially identify with the same topic along with supplement classification information by AIS. There is lack of evidence that leads to last statement “However, there are few studies that focus solely on supplementation in the younger and female population and no definitive conclusions can be drawn.” Do the authors suggest there is sufficient evidence in male population? If so need to highlight this.

Response of the authors: In accordance with the reviewer's suggestion, the introduction has been modified by adding more information in order to be more relevant. In addition, the highlighted sentence has been modified.

  1. Page 3 – Recommend moving descriptive data (participant characteristics) to the results section. Under the participants section, describe participants in detail and include inclusion and exclusion criteria.

Response of the authors: Although in previous studies published in mdpi we have included descriptive data in materials and methods, we accept the reviewer's suggestion, as well as expanded in the materials and methods section.

  1. Page 4 – Results section can begin with descriptive data highlighting participant characteristics.

Response of the authors: Although in previous studies published in mdpi we have included descriptive data in materials and methods, we accept the reviewer's suggestion.

  1. Page 6 – Reword “Table 3 shows the mean with its standard deviation and the ANOVA of the age, training days and SS consumed of the segmented sample according to playing position. No significant difference was found in any of the variables analyzed (F = 0.342 - 2.424; p = 0.795 - 0.074).” The above statement can be reworded to incorporate more details. AS it, represents a table/figure legend.

Response of the authors: Following the reviewer's suggestions, additional information has been added.

  1. Page 10: Discussion – Recommended replacing “unintentional doping” to a more scientific terminology. Or simple stating “ There is risk of adverse effect”

Response of the authors: Thank you for the comment. The sentence has been modified.

  1. Page 10: “However, the nutritionist is the main person in charge of advising on the consumption of SS in the sample of our soccer players, with a presence of 77.1% combining the club's internal nutritionist (65.7%) as well as external ones (11.4%)” Statement is unclear. Is the nutritionist promoting SS use or advising about SS consumption in general? Clarity is needed on this statement.

Response of the authors: Thank you for the comment. The club's nutritionist advises players on which supplements may be useful and which may be inadvisable or unnecessary, as well as examining the brand of products players intend to consume. In turn, the club itself has a partnership with a supplementation brand, so in this case, depending on the availability of the products and the economy, the nutritionist is in charge of scheduling supplements of that brand among the players. In short, the nutritionist is in charge of including, modifying or reducing the consumption of supplements depending on the competitive category, gender and playing position.

  1. The statement that follows is also unclear – “This is important, as this is one of the figures suitable for recommending the consumption of supplements, as they have scientific knowledge in the field of nutrition and supplementation, which implies a better choice by athletes when opting to consume some SS or others in terms of their level of scientific evidence and safety [16–18].” Clarity is needed – whether nutritionist is advising on safe choices or confirming SS use with Athletes. Was education piece assessed in this study? Results were not reported for this. The statement is very vague.

Response of the authors: Following the reviewer's suggestions, the sentence has been modified.

  1. Since hypotheses were provided earlier, the discussion section can begin if these were accepted or rejected. This way it becomes easier for the reader to understand the discussion points. 

Response of the authors: Thank you for the comment. The beginning of the discussion has been modified in response to the three hypotheses mentioned at the end of the introduction in order to improve reading comprehension.

Comments on the Quality of English Language

There is an average command of English language in general. Some sentences seem very vague especially in the discussion section. The results section is very brief and discussion points do not align well with the results. Overall there is extensive use of contrast and prepositions though out the text making the paper come across very layman like as opposed to scientific writing. Would recommend proofreading the whole article and revising as needed to improve the quality.

Response of the authors: We have revised the article again to optimize the translation.

Reviewer 2 Report

Comments and Suggestions for Authors

P2: the required sample size was 70 athletes: I think the sample size is not enough to be relied upon or generalized.

P3: Table 1. Descriptive data on age and training days as a function of sex, category and playing positions: I think table 1. Should be in results section.

P9: Table 4: I suggest to give a brief about each component of supplements of (A, B, and C) group in the introduction

Some discussion paragraphs should be moved into the introduction, as I suggest that authors have the ability to improve the discussion and focus on the results and their justifications.

Comments on the Quality of English Language

Minor editing of English language required

Author Response

P2: the required sample size was 70 athletes: I think the sample size is not enough to be relied upon or generalized.

Response of the authors: We are aware of the limitation of the sample size, for this reason we have expanded the information on this issue in the limitations section, although we emphasize that our sample is significant in this type of population and was used according to the statistical principles applied (2.1. section).

P3: Table 1. Descriptive data on age and training days as a function of sex, category and playing positions: I think table 1. Should be in results section.

 Response of the authors: Although in previous studies published in mdpi we have included descriptive data in materials and methods, we accept the reviewer's suggestion.

P9: Table 4: I suggest to give a brief about each component of supplements of (A, B, and C) group in the introduction

 Response of the authors: Thank you for the comment. We have added more information in the introduction regarding the description of each category, including some examples of each. However, we think it is favorable to keep the introduction brief and concise in order to provide more detail in the discussion (in the discussion we detail all the supplements that stood out based on the statistics).

Some discussion paragraphs should be moved into the introduction, as I suggest that authors have the ability to improve the discussion and focus on the results and their justifications.

Response of the authors: Following the reviewer's suggestions, the introduction has been optimized by adding information and trimming the discussion to improve readability.

Reviewer 3 Report

Comments and Suggestions for Authors

The objectives of the current study were to describe the consumption of sports supplements (CSS) in elite soccer players and asses its association with their competitive level, playing position and sex. This study can be accepted in the current form after addressing the major issues below.

1.      The practical application and the benefits of this study should be added in the abstract along with the last line conclusions

2.      the calculation of sample size was not clear and it is very small (70 players).

3.       the rationale of conducting this study was not clear in the abstract and introduction.

4.      How the tool used (questionnaire) was validated? It is not clear

5.        English version of the questionnaire should be provided as supplementary materials

6.      Some type errors should be addressed such as in the abstract “males' and females'.”

7.       The results section is well presented, however, avoidance of results repeating in the discussion section is necessary.  

8.      High percentage of self-citation was noticed.

Comments on the Quality of English Language

Minor editing of English language required

Author Response

The objectives of the current study were to describe the consumption of sports supplements (CSS) in elite soccer players and asses its association with their competitive level, playing position and sex. This study can be accepted in the current form after addressing the major issues below.

  1. The practical application and the benefits of this study should be added in the abstract along with the last line conclusions

Response of the authors: Thank you for the comment. We have modified the abstract by adding information related to practical applications while respecting the maximum word limit.

  1. the calculation of sample size was not clear and it is very small (70 players).

Response of the authors: We are aware of the limitation of the sample size, for this reason we have expanded the information on this issue in the limitations section, although we emphasize that our sample is significant in this type of population and was used according to the statistical principles applied (2.1. section).

  1. the rationale of conducting this study was not clear in the abstract and introduction.

Response of the authors: We have added information in the introduction to enhance the justification for this study.

  1. How the tool used (questionnaire) was validated? It is not clear

Response of the authors: The information related to the validation of the questionnaire has been described in greater depth (2.4. section), in addition to adding possible limitations in the discussion section (4.2. section).

  1. English version of the questionnaire should be provided as supplementary materials

Response of the authors: We appreciate the reviewer's comment but the authors cannot provide the English version of the questionnaire because it does not exist and has not been validated. To carry out this process, the guidelines for the translation and cultural adaptation of health measurement instruments should be followed.

https://doi.org/10.1016/j.jclinepi.2014.11.021

  1. Some type errors should be addressed such as in the abstract “males' and females'.”

Response of the authors: Thank you for the comment. Grammatical errors have been corrected.

  1. The results section is well presented, however, avoidance of results repeating in the discussion section is necessary.  

Response of the authors: Following the reviewer's suggestions, we have revised the discussion to reduce unnecessary repetition of results.

  1. High percentage of self-citation was noticed.

Response of the authors: The authors have made a modification to the self-citations although we must highlight that there are several lines and research projects that are being developed on food and supplement consumption in the sports field. This has an impact on the increase in the number of publications on the subject, which until now was limited.

Round 2

Reviewer 1 Report

Comments and Suggestions for Authors

None

Reviewer 3 Report

Comments and Suggestions for Authors

The manuscript is significantly improved and can be accepted in the current form